# TDRD5 Is Required for Spermatogenesis and Oogenesis in *Locusta migratoria*

**DOI:** 10.3390/insects13030227

**Published:** 2022-02-24

**Authors:** Sufang Deng, Junxiu Wang, Enbo Ma, Jianzhen Zhang, Shuping Xing

**Affiliations:** 1Research Institute of Applied Biology, Shanxi University, Taiyuan 030006, China; dengsufang@126.com (S.D.); wangjx0419@163.com (J.W.); maenbo2003@sxu.edu.cn (E.M.); zjz@sxu.edu.cn (J.Z.); 2College of Life Science, Shanxi University, Taiyuan 030006, China; 3College of Biological Sciences and Technology, Jinzhong University, Jinzhong 030600, China

**Keywords:** TDRD5, spermatogenesis, oocyte development, vitellogenesis, *Locusta migratoria*

## Abstract

**Simple Summary:**

Tudor domain-containing proteins (TDRDs) are a group of evolutionarily conserved and germline-enriched proteins, and most of them function in reproduction and the P-element-induced wimpy testis-interacting RNA (piRNA) pathway. We investigated LmTDRD5, a TDRD5 ortholog in *Locusta migratoria*. In males, *LmTdrd5* knockdown delayed meiosis phase transition and reduced the number of elongated spermatids and sperm count. The expression levels of two haploid germ cell marker genes, *LmCREM* and *LmACT,* as well as the sperm tail marker gene *Lm**Qrich2* were downregulated. In females, *LmTdrd5* knockdown decreased vitellogenin (*Vg*) and *Vg receptor* (*VgR*) transcript levels, thereby affecting ovarian development and oocyte maturation. Therefore, *LmTdrd5* may play an important role in locust reproduction, indicative of a conserved primary function of TDRD5.

**Abstract:**

Tudor family proteins exist in all eukaryotic organisms and play a role in many cellular processes by recognizing and binding to proteins with methylated arginine or lysine residues. TDRD5, a member of Tudor domain-containing proteins (TDRDs), has been implicated in the P-element-induced wimpy testis-interacting RNA (piRNA) pathway and germ cell development in some model species, but little is known about its function in other species. Therefore, we identified and characterized LmTDRD5, the TDRD5 ortholog in *Locusta migratoria*, a hemimetabolous pest. The *LmTdrd5* gene has 19 exons that encode a protein possessing a single copy of the Tudor domain and three LOTUS domains at its N-terminus. qRT-PCR analysis revealed a high *LmTdrd5* expression level in genital glands. Using RNA interference, *LmTdrd5* knockdown in males led to a lag in meiosis phase transition, decreased spermatid elongation and sperm production, and downregulated the expression of the two germ cell-specific transcription factors, *LmCREM* and *LmACT,* as well as the sperm tail marker gene *Lm**Qrich2.*
*LmTdrd5* knockdown in females reduced the expression levels of vitellogenin (*Vg*) and *Vg receptor* (*VgR*) and impaired ovarian development and oocyte maturation, thus decreasing the hatchability rate. These results demonstrate that *LmTdrd5* is essential for germ cell development and fertility in locusts, indicating a conserved function for TDRD5.

## 1. Introduction

Living organisms continue their species by reproduction, the way an individual produces offspring. The two main types of reproduction are asexual and sexual reproduction. In asexual reproduction, the offspring is produced from only one parent without fertilization, such as budding in yeast, regeneration in hydra and planaria, and fragmentation in blackworms. In sexual reproduction, the dominant way in nature, two parents produce offspring, and it requires a sperm to fertilize an egg, which forms a zygote [1,2]. Insects are a diverse group with a huge population, including prolific agricultural pests. Insect reproduction has broad significance in basic biology and potential application in biological control. Like many animals, most insect species reproduce sexually through several events, from mating to oviposition, and complex processes in each sex, including spermatogenesis in the testis and oogenesis in the ovary. Both spermatogenesis and oogenesis have been extensively studied in the model insect *Drosophila melanogaster*. Many genes and signaling pathways are revealed in these processes, and they advance our understanding of the underlying molecular mechanisms [3,4,5,6,7,8,9,10,11,12,13,14,15,16,17]. However, in other species, the information is fragmented.

*Locusta migratoria*, a devastating agricultural pest, causes locust plagues resulting in serious economic losses and shortage of food for both people and livestock in the area [18]. Therefore, keeping their population under control is crucial. *L.**migratoria*, from Orthoptera, has been used as a hemimetabolous model since its genome was sequenced in 2014 [19]. Recent studies focus on ovarian development and oocyte maturation to understand the molecular mechanism of female reproduction in this species [20,21,22,23,24]. Although a few studies have been conducted on the cellular biology of spermatogenesis, such as the fine structure and ultrastructure of sperm [25,26,27], the genes controlling spermatogenesis or male reproduction in this species remain largely elusive.

Tudor proteins are members of a royal superfamily that also contains the Chromo, MBT, PWWP, and plant Agenet domain proteins [28,29]. Since the first Tudor protein discovered in *D. melanogaster* [30], these proteins are also found in mouse, human, zebrafish, yeast, and plants [31,32,33,34,35,36,37,38,39,40,41]. Tudor proteins bind to proteins with methylated arginine or lysine residues and take part in cellular processes, such as DNA methylation, RNA metabolism, transcription regulation, DNA damage detection and repair, the P-element-induced wimpy testis (PIWI)-interacting RNA (piRNA) pathway, and the maintenance of genomic stability [32,42]. In particular, a special group of Tudor domain-containing proteins (TDRDs) is highly expressed in the gonads and selectively bind methylarginine, implicating a function for these proteins in gametogenesis [43]. Ten TDRD proteins have been identified in mice, and most of them have functions in spermatogenesis and male fertility [32,44,45,46]. For example, TDRD6 interacts directly with MILI and MIWI, the components of the chromatoid body (CB) and the piRNA pathway, and is required for spermatogenesis. *Tdrd6*-knockout mice did not produce elongated spermatids but round spermatids with defective CBs [47]. TDRD7 is also essential for CB assembly and spermatid development, but the underlying mechanism might differ from TDRD6, in which TDRD7 suppressed the retrotransposon *LINE1* but TDRD6 did not [45]. Several studies focusing on TDRD5 have revealed that it is a component of intermitochondrial cement (IMC) and CB and is an important factor for piRNA biogenesis [48,49,50,51,52]. *Tdrd5*-deficient males were sterile because of defective spermiogenesis caused by disorganized IMCs and CBs, in which the key components TDRD1/6/7/9 and MILI/MIWI/MIWI2 were mislocalized. TDRD proteins are evolutionarily conserved and also found in other species. In *D. melanogaster*, the TDRD5 ortholog Tejas is localized to the nuage, the counterpart of CB in mammals, and required for germline piRNA production and retroelement repression [9]. Loss of function of Tejas led to sterile females but fertile males, indicating a role of Tejas in oogenesis [9].

In this study, we identified a TDRD5 ortholog in *L. migratoria*, named LmTDRD5, and addressed the question of whether LmTDRD5 plays a role in spermatogenesis and oogenesis. By using RNA interference (RNAi) technology, we showed that *LmTdrd5* knockdown in males decreased the number of elongated spermatids and eventually reduced the sperm count, whereas *LmTdrd5* knockdown in females caused defective ovarian development and oocyte maturation. These data demonstrate that LmTDRD5 is essential for both spermatogenesis and oogenesis in locusts.

## 2. Materials and Methods

### 2.1. Insects

*L. migratoria* eggs were obtained from the Insect Protein Co., Ltd., Cangzhou, Hebei, China, and were incubated in sandy soil covered with a clingfilm at 30 ± 1 °C. After hatching, the nymphs were fed with fresh wheat sprouts and oat meal in a climate-controlled cabinet (Yiheng, Shanghai, China) with a light–dark cycle of 14 h:10 h and 40 ± 10% relative humidity at 30 ± 1 °C. Instar nymphs at different stages and adult locusts were used in this study.

### 2.2. Isolation of LmTdrd5 Gene and Bioinformatics Analysis

Tejas, the TDRD5 ortholog in *D. melanogaste*r, was chosen as a query for local BLAST searches of transcriptome data of *L. migratoria* [53]. The obtained cDNA was sequence confirmed and then blasted in the *Locusta migratoria* genome at NCBI to obtain the gene structure [19]. To predict the open reading frame (ORF) and physicochemical properties of *LmTdrd5*, the EXPASy online tools (http://www.expasy.org/tools/dna.html (accessed on 20 May 2020)) were used. For analysis of the domain structure, SMART (http://smart.embl-heidelberg.de/ (accessed on 10 August 2021)) was applied.

### 2.3. RNA Preparation and Real-Time Quantitative Reverse Transcription PCR (qRT-PCR)

Whole bodies at different developmental stages, from egg to adult, and tissues from the foregut (FG), midgut (MG), hindgut (HG), integument (IN), Malpighian tubule (MT), gastric caecum (GC), fat body (FB), testis (TE), and ovary (OV) of the fifth-instar nymphs at day 3 (N5D3) were sampled and frozen immediately in liquid nitrogen and stored at −80 °C for total RNA preparation. Samples from three replicates were collected.

Total RNA was extracted with TRIZOL reagent (Invitrogen, Carlsbad, CA, USA). First-strand cDNA was synthesized using Reverse Transcriptase M-MLV (TaKaRa Biotechnology, Co., Ltd., Shiga, Japan). qRT-PCR was performed in a Light Cycler480 (Roche Applied Science, Mannheim, Germany) with a protocol of 94 °C for 2 min, followed by two-step PCR, 40 cycles at 94 °C for 15 s, and at 60 °C for 31 s. Each PCR assay contained 10 μL of 2× SYBR^®^ qPCR MasterMix, 0.8 μL of primers (10 μM), and 4 μL cDNA template in a final volume of 20 µL. Two technical repeats were performed. β-actin was used as an internal control. The primers used are listed in Appendix A. The 2^−^^△△^^C^^t^ method was used to analyze the qRT-PCR data. Statistical analyses were performed by one-way analysis of variance, followed by Tukey’s honestly significant difference (HSD) test with SPSS v16.0 (IBM Corporation, Armonk, NY, USA).

### 2.4. Double-Stranded RNA (dsRNA) Synthesis and Injection

ds*LmTdrd5* and ds*GFP* were synthesized using T7 RiboMAX™ Express RNAi System (Promega, Madison, USA) as described previously [18]. The dsRNA was injected into the locust at the internode membrane of the second and third abdominal segments by using an injection needle. The injection was administered on the first day of the fifth-instar nymphs (N5D1), then at N5D4, on the first day of adult (AD1), and at AD4, with 12 μg dsRNA per nymph and 15 μg per adult, respectively.

### 2.5. Histology and Microscopy

Testes of locusts from N5D3, N5D5, AD2, AD5, and AD6 were dissected in phosphate buffered saline (PBS) and fixed in 4% paraformaldehyde fixative. To prepare the slide, the seminiferous tubules were randomly selected from the fixed testis, rinsed three times with PBS, stained with improved phenol fuchsin stain (Yuanye Bio-technology Co., Ltd., Shanghai, China) for 10–15 min at room temperature, and squashed under a coverslip. All images were taken under a microscope (Leica Microsystems, Wetzlar, Germany).

Ovaries of adult female locusts at 4, 6, and 8 days post-adult eclosion (PAE) were excised in PBS. After the outer membrane was removed, the size of ovarioles and primary oocytes was measured and photographed under a Leica M205C microscope. The data for primary oocyte size were analyzed by the two-sample and two-tailed *t*-test and presented as a scatter dot plot (prepared by GraphPad Prism 8). All differences were considered significant at *p* < 0.05.

### 2.6. Analysis of Hatchability

To investigate the hatchability, three types of combinations were prepared. In type I, there were 30 ds*LmTdrd5-*treated or 30 ds*GFP-*treated males with 30 wild type (WT) females. In type II, there were 30 ds*LmTdrd5-*treated or 30 ds*GFP-*treated females paired with 30 WT males. In type III, there were 30 males and 30 females injected with ds*LmTdrd5*. The control is the equal number of males and females injected with ds*GFP*. Each combination group of adult locusts was incubated in a cage with moist sandy soil under the same conditions mentioned in Section 2.1. After mating behavior was observed, the egg pods were collected every other day from each group until all adults stopped mating.

Egg pods were kept for 10 days in bottles filled with moist soil at room temperature. Eggs were mixed from the egg pods of the same group. One hundred eggs were randomly selected from the same group and were incubated in a container with sterile, moist sandy soil. The first-instar nymphs in the containers were counted daily until no nymphs appeared. The hatchability rate was calculated as a percentage, i.e., the number of nymphs from 100 eggs. Five counting processes were repeated for each group.

## 3. Results

### 3.1. LmTDRD5 Is a LOTUS-Containing Protein Highly Expressed in the Testes and Ovaries

To identify the *LmTdrd5* gene, Tejas, the TDRD5 ortholog in *D. melanogaster*, was used as a query to perform local BLAST searches of transcriptome data of *L. migratoria* [53]. A 4749-bp sequence was obtained and further cloned by RT-PCR. This sequence was then used to blast the *L. migratoria* genome at NCBI (taxid:7004). The *LmTdrd5* gene consists of 20 exons and 19 introns (Figure 1a). The cDNA included an 18-bp 5′ untranslated region (UTR) and a 4731-bp ORF encoding 1577 amino acid residues (GenBank accession no. OM 324402). The theoretical molecular weight was 179.20 kDa, and the pI was 5.10. Domain architecture analysis by SMART showed that LmTDRD5 had four conserved domains, namely one single copy of the Tudor domain and three LOTUS domains at its N-terminus (Figure 1b).

To obtain the LmTdrd5 expression profile, a series of qRT-PCR assays was performed in various tissues at different developmental stages. The results showed that LmTdrd5 was expressed in all examined stages, from egg to adult, with a peak at the fifth-instar nymphs (Figure 1c). Interestingly, LmTdrd5 was most highly expressed in the testes and ovaries (Figure 1d), suggesting a role of this gene in the reproductive phase.

### 3.2. LmTdrd5 Knockdown Reduced the Hatchability Rate in Female Locusts

To test the role of *LmTdrd5* in the reproductive phase, we investigated the hatchability rate for these ds*LmTdrd5*- or ds*GFP*-treated locusts in three types of combination groups (100 eggs were counted in each group). In type I, only males were dsRNA-treated and females were wild type (WT). The hatchability rate in the ds*LmTdrd5*-treated group was 60.1%, a significant reduction compared with 93% in the control ds*GFP*-treated group (Figure 2a), indicative of male fertility defect. In type II, only females were injected with dsRNA, and males were WT. In this combination, the rate from ds*Lm**Tdrd5*-treated group reached 73.1%, still lower than 91% in the control (Figure 2b), suggesting female fertility defect. In type III, both males and females were treated with dsRNA. Interestingly, the rate of ds*LmTdrd5*-treated males and females was 45%, much lower than the control 91% (Figure 2c). These data suggest reproductive defects in both males and females.

### 3.3. DsLmTdrd5-Treated Males Produced a Lower Number of Sperms

To find the reproductive defect in ds*Lm**Tdrd5*-treated males, the testes from these fifth-instar nymphs and adult locusts were dissected. The testis size was not affected by the ds*LmTdrd5* injection (Appendix A). We focused on the transition zone of seminiferous tubules in the stained testes. An obvious difference was observed between ds*LmTdrd5*-treated and ds*GFP*-treated males: the meiotic phase lagged in the ds*LmTdrd5*-treated group compared with the ds*GFP*-treated group. For example, ds*GFP*-treated males were already in metaphase II but ds*LmTdrd5*-treated males were still in prophase II (Figure 3a,b). Later, many spermatids were formed and elongated in ds*GFP*-treated males, whereas only a portion of spermatids was elongated (others were still in the round spermatid stage) and the spermatid count was reduced by almost half in ds*LmTdrd5*-treated males (Figure 3c,d). In the advanced developmental stage, bunches of the needle-formed sperms were observed in the ds*GFP*-treated males but the round spermatids were still found in ds*LmTdrd5*-treated males (Figure 3e,f). At least one-third of round spermatids did not elongate. Therefore, *LmTdrd5* may be involved in meiosis, spermatid elongation, and sperm maturation in male locusts.

In the mouse *Tdrd5^−^*^/−^ mutant, the expression level of *ACT*, an activator of *CREM* (a germ cell-specific transcription factor involved in the cAMP-dependent signaling pathway), was significantly reduced in the testis [51]. Because of phenotypic similarities in spermatogenesis between the mouse *Tdrd5^−^*^/−^ mutant and *LmTdrd5* knockdown, we examined *LmCREM* and *LmACT* expression levels in *LmTdrd5* knockdown. Our qRT-PCR results indicated a significant decrease in *LmCREM* and *LmACT* expression in ds*LmTdrd5*-treated locusts at N5D6, ADD4, and ADD6 (Figure 3g,h; Appendix A). In addition, we examined a sperm tail marker gene, *glutamine*
*rich 2* (*Qrich2*) [54], and found that the expression level of its ortholog in *L. migratoria*
*LmQrich2* was suppressed by 13%, 21%, and 34% at N5D6, ADD4, and ADD6, respectively (Figure 3i; Appendix A). Therefore, *LmTdrd5* plays a role in spermatogenesis.

### 3.4. LmTdrd5 Knockdown Impaired Ovarian Development and Oocyte Maturation

To investigate the role of *LmTdrd5* in female reproduction, the same amount of dsRNA as used in males was used in females. We dissected ovaries from these dsRNA-treated locusts and focused on the phenotype of ovarian development and oocyte maturation. At four days PAE (ADD4), almost all ovaries in ds*Lm**Tdrd5*-treated locusts were smaller than those in the ds*GFP*-treated locusts, and the primary oocytes were also remarkably smaller (Figure 4a). At six days PAE (ADD6), the ds*Lm**Tdrd5*-treated females produced two types of ovaries, i.e., two levels of retarded ovarian development, and primary oocyte maturation with smaller primary oocyte length. Type I: 70% of ds*Lm**Tdrd5-*treated females had yellow ovaries and primary oocytes, which were thinner and shorter than those of the controls. Type II: the rest (30%) showed a serious phenotype with small ovaries and white primary oocytes, which appeared at a standstill (Figure 4b). On average, the primary oocyte length in types I and II was 3.64 mm and 1.34 mm, respectively (Figure 4c). Both were significantly reduced compared with the control ds*GFP*-treated locusts (5.77 mm) (Figure 4d). At eight days PAE (ADD8), the primary oocytes in the control ovaries matured normally and those in the type I ovaries were at the stage of reaching maturity, although a bit slower than those in the control. However, the primary oocytes were still underdeveloped in type II (Figure 4c).

### 3.5. Vg and VgR Expression Levels Were Affected in dsLmTdrd5-Treated Females

Vitellogenin (Vg) and vitellogenin receptor (VgR) are two key factors for oocyte development and maturation in insects. To explain the reason why *LmTdrd5* knockdown led to defective oocyte development and maturation, we examined the expression levels of *Lm**VgA* and *Lm**VgB* as well as *Lm**VgR* in ds*LmTdrd5*-treated female locusts. qRT-PCR data showed 88.2% and 86.3% reductions in *Lm**VgA* and *Lm**VgB* expression levels, respectively, at ADD4, whereas the *Lm**VgR* expression level decreased only 10.3% (Figure 5a–c). At ADD6 and ADD8, the ds*LmTdrd5*-treated females had two types of ovaries. For type I females, the relative expression levels of *Lm**VgA* and *Lm**VgB* in fat bodies more or less reached the level of the control (Figure 5d,e). However, the *Lm**VgR* expression level was only 21.1% of that in the control at ADD6 and reached 83.7% at ADD8 (Figure 5f). For type II, ds*LmTdrd5* treatment reduced the *Lm**VgA* expression level in the control to 17.5% and 33.9% at ADD6 and ADD8, respectively (Figure 5g). Similarly, the *Lm**VgB* expression level reduced to 20.2% and 35.1% (Figure 5h). However, the *LmVgR* expression level differed from that in type I, with a significant reduction (54%) only found at ADD8 (Figure 5i). Therefore, *LmTdrd5* might regulate ovarian development and oocyte maturation by mediating in *LmVg* and *LmVgR* expression.

## 4. Discussion

We identified and characterized *LmTdrd5*, the *Tdrd5* ortholog, in *L. migratoria*. Our data showed that LmTDRD5 had evolutionarily conserved domains and functions, implying that it may share the same or similar molecular mechanism as other TDRD5s to exert its function.

Like mouse TDRD5, LmTDRD5 possesses only one repeat of the Tudor domain with three LOTUS domains at its N-terminus. This Tudor domain in mouse TDRD5 is an extended Tudor domain (eTudor/eTud) containing 180 amino acids that consists of a canonical Tudor domain (60 amino acids), α-helix linker, and a staphylococcal nuclease-like domain. These three structural elements are required for TDRD5 binding to the PIWI proteins [55,56,57]. eTudor domains are also found in other TDRD proteins, such as the 36 eTudor domains present in the 12 human TDRD proteins. These eTudor domains specifically recognize the RA/RG repeats in the N-termini of PIWI proteins, and symmetric arginine demethylation of RA/RG is essential for binding to them [55]. Although the Smart program recently recognized the Tudor domain as a canonical domain in LmTDRD5, considering the sequence and topology similarity between LmTDRD5 and mouse TDRD5, this single Tudor domain in LmTDRD5 might also be an eTudor domain. When we used the CDD Search program on the NCBI website (https://www.ncbi.nlm.nih.gov/structure/cdd/ (accessed on 15 December 2021)), it revealed a large Tudor domain of 129 amino acids in LmTDRD5, supporting our assumption. LmTDRD5 could bind PIWI proteins by the eTudor domain. We further examined the N-termini of PIWI proteins in *L. migratoria* and they all contained RA/RG repeats (Appendix A), indicating that this feature of PIWI proteins is also conserved.

The LOTUS domain (also known as OST-HTH) is an RNA-binding domain found in germline proteins, such as Oskar, TDRD5, TDRD7, and MARF1 (meiosis arrest female 1), and is mainly associated with the small RNA pathway [58,59,60]. This domain comprises 80–100 amino acids that form a winged helix-turn-helix fold. The LOTUS domain is conserved in bacteria, fungi, and plants but experimentally studied only in animals [60]. The LOTUS domain selectively binds to G-rich RNAs and the RNA G-quadruplex (G4) secondary structure in vitro [61]. However, whether the LOTUS domain in LmTDRD5 has the same specificity needs further investigation.

Mouse TDRD5 can associate with pachytene piRNA precursors by its LOTUS domain and is thus essential for pachytene piRNA biogenesis [50,62], which contains a low level of transposon sequences, suggesting that they regulate mRNA and long non-coding RNA in mouse testis [63,64,65] and are required for spermatogenesis.

In addition, in the mouse *tdrd5*-deficient mutant, the expression of *ACT*, which encodes a coactivator of the germline transcription factor *CREM*, was greatly reduced in the round spermatids, and the expression of *CREM* target genes, such as *Prm1*, *Prm2*, *Tnp1*, *Calspermin*, and *RT7*, was also reduced [51,66,67,68], but not the expression of *CREM* itself. Mouse TDRD5 may thus function independently of piRNA in posttranscriptional gene regulation, and defective spermatogenesis in the mouse *tdrd5^−/−^* mutant may be caused by the downregulation of these *CREM* target genes. Intriguingly, *LmTdrd5* knockdown led to downregulated orthologs of both *ACT* and *CREM* in the testis of *L. migratoria* (Figure 3g,h), implying that LmTDRD5 may also have a function in posttranscriptional gene regulation. However, we cannot conclude that the *LmTdrd5* knockdown phenotype in males was indeed caused by the downregulation of *ACT* and *CREM*.

Tejas and Tapas, the orthologs of TDRD5 and TDRD7 in *D. melanogaster*, have also been implicated in the piRNA pathway, and both interact with PIWI proteins. However, the precise molecular mechanisms underlying the function of these two genes during oogenesis are unknown. Tejas and Tapas bind to germline protein Vasa by their LOTUS domains [68]. Vasa is located in the nuage, the counterpart of CB in mammals, and involved in the piRNA pathway. Thus, Tejas and Tapas might form a complex with Vasa and thereby function in piRNA-mediated transcript repression [68]. In the spermatogenesis of *L. migratoria*, CBs appeared in the spermatids [26]. Thus, LmTDRD5 accumulated in CBs may also bind to Vasa and function in the piRNA pathway. Like Tejas, the detailed molecular mechanism of LmTDRD5 in locust reproduction requires further investigation.

## 5. Conclusions

We characterized LmTDRD5, the TDRD5 ortholog in *L. migratoria*. *LmTdrd5* was highly expressed in the testes and ovaries. *LmTdrd5* knockdown in males delayed meiosis phase transition, impaired spermatid elongation, and eventually reduced the sperm count. In line with these phenotypes, the expression levels of the two haploid germline marker genes, *LmCREM* and its activator *LmACT,* as well as the sperm tail marker gene *Lm**Qrich2* were reduced. *LmTdrd5* knockdown in females decreased *Lm**Vg* and *LmVgR* expression and blocked ovarian development and oocyte maturation. In short, *LmTdrd5* plays an important role in locust reproduction, both spermatogenesis and oogenesis, indicative of a conserved function in TDRD5 family members. However, this study provides only some information on *LmTdrd5* and the molecular mechanism by which it exerts its function remains to be answered.

## Figures and Tables

**Figure 1 insects-13-00227-f001:**
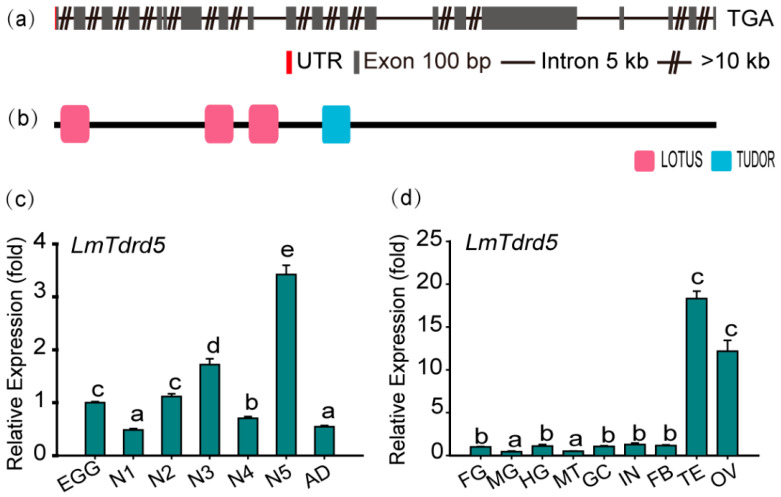
Gene structure and expression profile of *LmTdrd5.* (**a**) Schematic representation of the exon–intron structure of *LmTdrd5.* Exons contributed to the coding sequence (CDS) marked in goldenrod, red represents the untranslated region (UTR), the black line denotes an intron, and the double slashes indicate a break of introns. Gene structures were deduced from the analysis on the *L. migratoria* genome database (http://www.locustmine.org/index.html (accessed on 12 September 2021)). (**b**) LmTDRD5 was analyzed by SMART (Smart.embl-heidelberg.de), showing the four conserved domains marked in two colors. The first three in pink are the LOTUS domains, and the TUDOR domain is in blue. (**c**) *LmTdrd5* expression at different locust developmental stages. Egg (EGG), first-instar (N1), second-instar (N2), third-instar (N3), fourth-instar (N4), fifth-instar (N5) nymphs, and adult (AD). (**d**) *LmTdrd5* mRNA expression in different tissues of fifth-instar nymphs. FG, foregut; MG, midgut; HG, hindgut; MT, Malpighian tubules; GC, gastric caecum; IN, integument; FB, fat body; TE, testis; OV, ovary. Letters on the bars represent significant differences among the developmental stages or different tissues based on Tukey’s HSD multiple comparison test (*p* < 0.05).

**Figure 2 insects-13-00227-f002:**
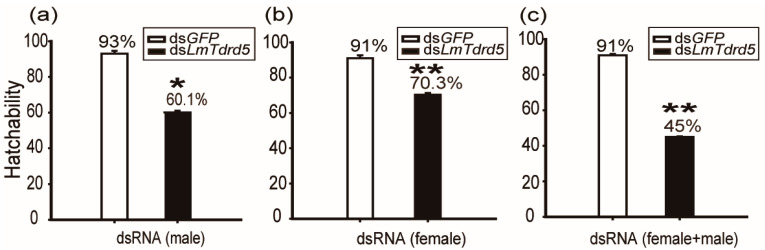
Analysis of hatchability. Hatchability was calculated as a percentage. (**a**) Ds*LmTdrd5* or ds*GFP* injected only in males. (**b**) Ds*LmTdrd5* or ds*GFP* injected only in females. (**c**) Both males and females injected with either ds*LmTdrd5* or ds*GFP*. Statistically significant differences are indicated as follows: * *p* < 0.05; ** *p* < 0.01.

**Figure 3 insects-13-00227-f003:**
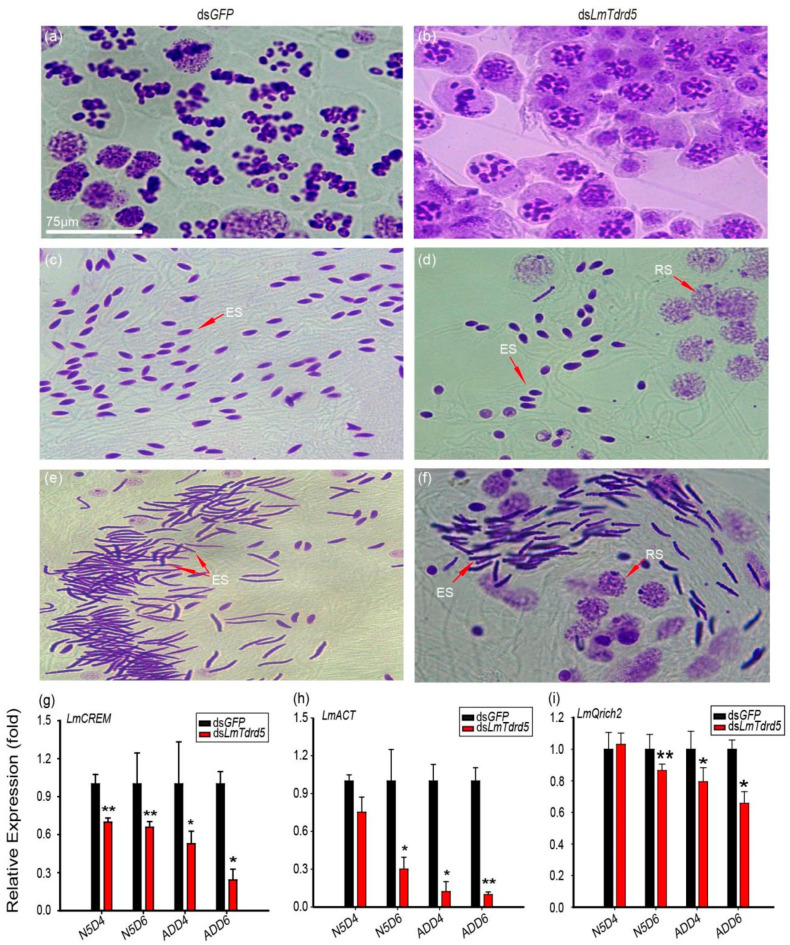
Defects in spermatogenesis of ds*LmTdrd5-*treated locusts. (**a**–**f**) Cytological phenotype of three representative stages during spermatogenesis observed at the transition zone of seminiferous tubules from testes stained with the improved phenol fuchsin stain. Purple represents chromatin. (**a**,**c**,**e**) Haploid germ cells in the meiotic II stage and elongated spermatids from ds*GFP*-treated testes. (**b**,**d**,**f**) Haploid germ cells in the meiotic II stage and elongated spermatids from ds*LmTdrd5*-treated testes. Meiosis phase transition was delayed, and the sperm count was significantly reduced. RS, round spermatids; ES, elongated spermatids. (**g**–**i**) Effect of ds*LmTdrd5* injection on *LmCREM* expression levels (**g**), *LmACT* (**h**), and *Lm****Qrich2*** (**i**) in the testes on days N5D4, N5D6, ADD4, and ADD6 compared with the control ds*GFP*-treated locusts. Data are presented as mean ± standard error of the mean (*n* = 9–15). Statistically significant differences are indicated as follows: * *p* < 0.05; ** *p* < 0.01.

**Figure 4 insects-13-00227-f004:**
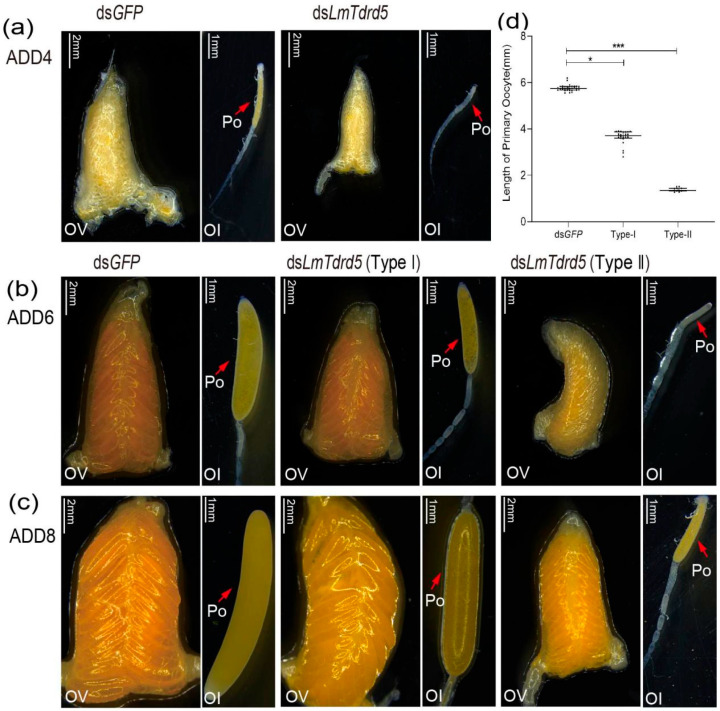
Effects of ds*LmTdrd5* injection on ovarian development and oocyte maturation. (**a**–**c**) Morphology of ovaries and ovarioles in female locusts after ds*GFP* or ds*LmTdrd5* treatment. Ol, ovariole; Ov, ovary; Po, primary oocyte. ADD4, adult of 4 days; ADD6, adult of 6 days; ADD8, adult of 8 days. Scale bars: Ov, 2 mm; Ol, 1 mm. (**d**) Primary oocyte length of locusts injected with ds*LmTdrd5* or ds*LmGFP* at ADD4, ADD6, ADD8. Data are presented as scatter dot plot. The horizontal lines depict the median with range. Statistically significant differences are indicated as follows: * *p* < 0.05; *** *p* < 0.001. *n* = 11–33.

**Figure 5 insects-13-00227-f005:**
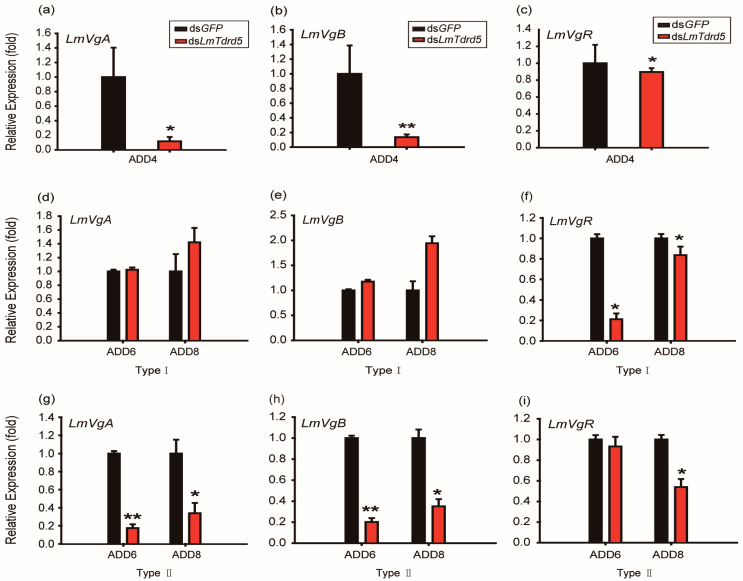
Effect of ds*LmTdrd5* injection on the transcript levels of vitellogenin (*Vg*) and its receptor. (**a**–**c**) Relative expression levels (fold) of *LmVgA* (**a**), *LmVgB* (**b**), and *LmVgR* (**c**) in the fat bodies and ovaries in locusts at ADD4. (**d**–**f**) Type I: Relative expression levels (fold) of *LmVgA* (**d**), *LmVgB* (**e**), and *LmVgR* (**f**) at ADD6 and ADD8. (**g**–**i**) Type II: Relative expression levels (fold) of *LmVgA* (**g**), *LmVgB* (**h**), and *LmVgR* (**i**) at ADD6 and ADD8. Data in all panels are presented as mean ± standard error of the mean (*n* = 9–15). Statistically significant differences are indicated as follows: * *p* < 0.05; ** *p* < 0.01.

## Data Availability

All data produced from this study are included in this published paper.

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
