# Peer review of "TDRD5 Is Required for Spermatogenesis and Oogenesis in Locusta migratoria"

_insects, 2022, doi:10.3390/insects13030227_

Round 1

Reviewer 1 Report

Tudor family proteins exist in all eukaryotic organisms. Authors at first characterized the TDRD5 ortholog from the migratory locust and demonstrate that LmTdrd5 is essential for germ cell development and fertility in locusts. All experiments are carefully done and the manuscript is well written. The results for males are of special interest because little is still known on the genes controlling spermtogenesis.

In my opinion, the manuscript needs only minor revision:

(1) The first part of the Introduction is too general and should be deleted

(2) In 2.5 authors should say here from which stage and age testes were dissected

(3) line 167: give only two decimals for the pI value

(4) Legend to Fig. 1a: I do not see any green color here

(5) Legend to Fig. 1b: the figure does not really show the "amino acid sequence"

(6) References: give all titles of papers in lowercase letters and all species names in italics

Author Response

Tudor family proteins exist in all eukaryotic organisms. Authors at first characterized the TDRD5 ortholog from the migratory locust and demonstrate that LmTdrd5 is essential for germ cell development and fertility in locusts. All experiments are carefully done and the manuscript is well written. The results for males are of special interest because little is still known on the genes controlling spermtogenesis.

Authors’ response: Thank you for your positive evaluation on our manuscript.

In my opinion, the manuscript needs only minor revision:

(1)The first part of the Introduction is too general and should be deleted

Authors’ response: Yes, we agree that first part of the introduction is a general information but it is necessary for the readers to understand logically the question we would like to address in this study.

(2)In 2.5 authors should say here from which stage and age testes were dissected

Authors’ response: The developmental stage of locust for testis dissection was added in the revised version.

(3)line 167: give only two decimals for the pI value

Authors’ response: Done in the revised version.

(4)Legend to Fig. 1a: I do not see any green color here

Authors’ response: Sorry for the mistake, it should be a red color here. It was changed in the revised version.

(5)Legend to Fig. 1b: the figure does not really show the "amino acid sequence"

Authors’ response: Yes, you are right, we did not show the amino acid sequence here. We thus deleted the "amino acid sequence of" and added “was” after LmTDRD5 in the revised version.

(6)References: give all titles of papers in lowercase letters and all species names in italics

Authors’ response: Done in the revised version.

Reviewer 2 Report

The authors reported a member of Tudor domain-containing (TDRD) proteins in the migratory locusts, the TDRD5 gene, to be specific. Its role in the regulation of locust reproduction was studied by applying some basic molecular techniques such as determination of expression profile by qPCR, phenotype study by RNAi and immunohistochemistry. It was found that knockdown of this gene led to the defects in spermatogenesis in the males and oogenesis in the females, which eventually resulted in low fertility of the insects. The reviewer has the following comments:

TDRD5 usually facilitates the production of piRNAs which maintains the germline RNA homeostasis through transposon silencing in many model systems. It is surprising to know in this manuscript that the expression of vitellogenin, one of the most abundant protein in oogenesis, as well as the receptor, is regulated by TDRD5, while a number of studies indicated direct involvement of juvenile hormone pathway for the activation of Vg transcription in locusts. The key factors known for the direct upregulation of Vg by hormonal pathway was not investigated in this assay. Are Vg and VgR regulated by TDRD5 in an indirect manner? What is the possible intermediate regulator in this species? Therefore, the reviewer is not completely convinced by the results in the current form of the manuscript in terms of the role of TDRD5 in females’ reproductive maturation.

The reviewer suggests that the alphabetic labels and other symbols in the histology images in Figure 3 be replaced for the purpose of a clearer indication.

Figure 4d, please use a scatter dot plot with the numbers of the insects used for the determination of the length in each group.

Line 246, remove “control”

Line 296, “consists of”

Line 303, delete “in”.

Author Response

The authors reported a member of Tudor domain-containing (TDRD) proteins in the migratory locusts, the TDRD5 gene, to be specific. Its role in the regulation of locust reproduction was studied by applying some basic molecular techniques such as determination of expression profile by qPCR, phenotype study by RNAi and immunohistochemistry. It was found that knockdown of this gene led to the defects in spermatogenesis in the males and oogenesis in the females, which eventually resulted in low fertility of the insects. The reviewer has the following comments:

Authors’ response: Thank you for your review. Your comments and suggestions will help us to improve the quality of our manuscript.

TDRD5 usually facilitates the production of piRNAs which maintains the germline RNA homeostasis through transposon silencing in many model systems. It is surprising to know in this manuscript that the expression of vitellogenin, one of the most abundant protein in oogenesis, as well as the receptor, is regulated by TDRD5, while a number of studies indicated direct involvement of juvenile hormone pathway for the activation of Vg transcription in locusts. The key factors known for the direct upregulation of Vg by hormonal pathway was not investigated in this assay. Are Vg and VgR regulated by TDRD5 in an indirect manner? What is the possible intermediate regulator in this species? Therefore, the reviewer is not completely convinced by the results in the current form of the manuscript in terms of the role of TDRD5 in females’ reproductive maturation.

Authors’ response: TDRD5 as a conserved germline enriched Tudor domain containing protein regulates spermatogenesis or oogenesis via the piRNA/piRNA pathway. LmTDRD5 might use the same mechanism to exert its function in locust reproduction. Based on the phenotype observed in the dsLmTdrd5-treated female locust, we examined the expression levels of the LmVg and LmVgR, two key factors of vitellogenesis in locust. Our result showed that the expression of these two genes was decreased significantly in the dsLmTdrd5 knockdowns. We agree with you on that this regulation is probably indirect. It is intriguing that if we can find the intermediate factors in this regulation pathway. Many factors were reported to regulate the expression of LmVg, the juvenile hormone (JH)/JH pathway was extensively investigated in this species. The relationship of the LmTDRD5 and JH pathway in mediating vitellogenesis is a curious issue that would like to address.        

The reviewer suggests that the alphabetic labels and other symbols in the histology images in Figure 3 be replaced for the purpose of a clearer indication.

Authors’ response: We removed the “x” and the related arrows in Figure 3, other symbols were still maintained for a clearer description.

Figure 4d, please use a scatter dot plot with the numbers of the insects used for the determination of the length in each group.

Authors’ response: Done in the revised version.

Line 246, remove “control”

Authors’ response: Done in the revised version.

Line 296, “consists of”

Authors’ response: Done in the revised version.

Line 303, delete “in”.

Authors’ response: Done in the revised version.

Round 2

Reviewer 2 Report

The revised version is now acceptable for consideration of the publication on Insects journal, although the question remains of how this gene regulates the expression of Vg and VgR in locusts.